# Large self-assembled clathrin lattices spontaneously disassemble without sufficient adaptor proteins

**Si-Kao Guo**[1], **Alexander J. Sodt**[2], **Margaret E. Johnson**[1]*

**1** TC Jenkins Department of Biophysics, Johns Hopkins University, Baltimore, Maryland, United States of America, **2** Eunice Kennedy Shriver National Institute of Child Health and Human Development, National Institutes of Health, Bethesda, Maryland, United States of America

* margaret.johnson@jhu.edu

**Data Availability Statement:** All relevant data are within the manuscript and its Supporting Information files. Software and executable input

## Abstract

Clathrin-coated structures must assemble on cell membranes to internalize receptors, with the clathrin protein only linked to the membrane via adaptor proteins. These structures can grow surprisingly large, containing over 20 clathrin, yet they often fail to form productive vesicles, instead aborting and disassembling. We show that clathrin structures of this size can both form and disassemble spontaneously when adaptor protein availability is low, despite high abundance of clathrin. Here, we combine recent *in vitro* kinetic measurements with microscopic reaction-diffusion simulations and theory to differentiate mechanisms of stable vs unstable clathrin assembly on membranes. While *in vitro* conditions drive assembly of robust, stable lattices, we show that concentrations, geometry, and dimensional reduction in physiologic-like conditions do not support nucleation if only the key adaptor AP-2 is included, due to its insufficient abundance. Nucleation requires a stoichiometry of adaptor to clathrin that exceeds 1:1, meaning additional adaptor types are necessary to form lattices successfully and efficiently. We show that the critical nucleus contains ~25 clathrin, remarkably similar to sizes of the transient and abortive structures observed *in vivo*. Lastly, we quantify the cost of bending the membrane under our curved clathrin lattices using a continuum membrane model. We find that the cost of bending the membrane could be largely offset by the energetic benefit of forming curved rather than flat structures, with numbers comparable to experiments. Our model predicts how adaptor density can tune clathrin-coated structures from the transient to the stable, showing that active energy consumption is therefore not required for lattice disassembly or remodeling during growth, which is a critical advance towards predicting productive vesicle formation.

## Author summary

Stochastic self-assembly of clathrin-coated structures on the plasma membrane is essential for transport into cells. We show here that even with abundant clathrin available, robust nucleation and growth into stable structures on membranes is not possible without sufficient adaptor proteins. Our results thus provide quantitative justification for why

files for the models used here are provided open source at github.com/mjohn218/NERDSS.

**Funding:** M.E.J. gratefully acknowledges funding from a National Institutes of Health MIRA Award R35GM133644. A.J.S. is supported by the Intramural Research Program of the NIH. The funders had no role in study design, data collection and anaylsis, decision to publish, or preparation of the manuscript.

**Competing interests:** The authors have declared that no competing interests exist.

structures observed to form *in vivo* can still spontaneously disassemble over many seconds. Although clathrin disassembly after productive vesicle formation requires the work of ATPases, our study shows that active energy input is not necessary to control remodeling during growth. With parameterization against *in vitro* kinetics of assembly on membranes, our reaction-diffusion model provides a powerful and extensible tool for establishing determinants of productive assembly in cells.

## Introduction

Clathrin-mediated endocytosis (CME) is an essential pathway used by all eukaryotes for transport across the plasma membrane [1]. Viral infection and cancer can result from failures in cargo and receptor transport [2]. For proper receptor internalization, the trimeric clathrin proteins must assemble a hexagonal lattice on membranes (termed clathrin-coated structures) that ultimately bend the membrane into budded clathrin-coated vesicles [3]. However, the nascent clathrin-coated structures only proceed to productive vesicles about half the time [4], otherwise, they disassemble. These transient and abortive structures are observed to contain at least ~20 clathrin trimers [3], providing enough trimers to assemble multiple hexagonally looped subunits. Such a relatively large (~120 nm in diameter), persistent (several seconds) [4], and crosslinked structure may seem too stable to spontaneously disassemble (that is, without energetic input). Indeed, disassembly of *completed* clathrin-coated vesicles is known to require the energy-consuming work of an ATPase [5,6]. However, for *maturing* clathrin-coated structures, some experiments have observed the ATPase at maturation sites [7], while others have rarely observed it [8,9]. Additionally, although several clathrin associated proteins (such as adaptor proteins) have been shown to tune the frequency of disassembly in transition structures [3,4,10,11], the complexity due to the dozens of proteins that function in clathrin-mediated endocytosis makes it unclear by what mechanism each protein is tuning disassembly. Fundamental questions thus remain: are some clathrin-coated structures dynamic and unstable enough to spontaneously disassemble, and if so, what factors can control this stability? Addressing these outstanding questions will help to establish under what conditions clathrin-coated structures develop into productive vesicles. Here, our simulations of clathrin recruitment and assembly on membranes reproduce *in vitro* kinetic data, validating a model that then establishes how the key parameters of adaptor density, volume to area (*V/A*) ratio, and clathrin concentration impact the kinetics and critical nuclei of clathrin-coated structures on membranes.

Although *in vivo* experiments have provided detailed insight into the dynamics of clathrin coat nucleation and budding [12,13], the multi-component and nonequilibrium nature of the assembly means that many possible mechanisms could regulate disassembly. For example, lipid enzymes are known to be present during coat maturation with a potential to dynamically alter membrane composition [14,15], and the aforementioned ATPases could impact structural stability and dynamics [5,6]. Fortunately, clathrin-coated structures can be reconstituted *in vitro*, where they have been shown to assemble robustly on membranes with clathrin and a minimal set of components [16–18]. Specifically, coat formation requires clathrin, a cytosolic adaptor protein that localizes clathrin to the membrane, and membrane binding sites that localize adaptors to the membrane [16–18], which physiologically is the essential lipid PI(4,5)P$_2$. Because these experiments report on conditions that drive robust and stabilized lattices at equilibrium, however, they have not reported what stoichiometry of clathrin, adaptors, and membrane area determines the transition from unstable to stable clathrin coats. Further, although the *in vitro*

concentrations of the protein components are often comparable to *in vivo* reported values, the *V/A* ratio is typically orders of magnitude higher than in cells. The *V/A* ratio is important because when proteins in solution (3D) can localize to a 2D surface, they can exploit dimensional reduction to dramatically increase stability of membrane associated complexes in a way that is strongly influenced by *V/A* values [19]. Thus, it is not known if clathrin-coated structures can disassemble spontaneously or if energetic input is required.

Kinetic measurements have provided a powerful variable to assess both models and mechanisms of solution assembly in diverse filament forming [20,21], aggregating [22,23], and capsid forming systems [24]. With recent kinetic experiments, Sarkar and Pucaydil [25] tracked fluorescently tagged clathrin as it accumulated on membranes with adaptors attached. These experiments thus quantified timescales of clathrin localization to membranes under well-defined initial conditions that drove robust assembly, while previous biochemical experiments constrain the binding free energy of clathrin-clathrin [26] and clathrin-adaptor [27] interactions. To assess the size and lifetimes of stable nuclei as they might occur in the cell, models must mimic the lower *V/A* ratio of the cell, recreating these experiments *in silico* at new conditions that are challenging experimentally. An advantage of the type of microscopic spatial modeling used here is that instead of requiring experiments at many distinct concentrations in order to fit rate equations to the data [23], our explicit physical models tightly constrain the subset of rate constants that can quantify the data.

Modeling has played an important role in understanding principles of clathrin-coated assembly, although each model, including ours, must make tradeoffs. To position our model and study relative to others, we identify model elements or questions that are both included and excluded in published studies. Simulations based on spatial models of clathrin assembly in solution [28–32] have determined how the clathrin-adaptor energetics and stoichiometry can control cage formation [29], albeit without membrane localization or kinetics. Simulations of cage formation on membranes have characterized remodeling or cage structural changes, but without addressing dependence on adaptor stoichiometry, clathrin concentration, or volume to area ratio [33–37]. Thermodynamic models have predicted cage sizes in solution, without adaptors included [38,39], while stochastic 1D models have predicted cage lifetimes, but lack structure, stoichiometry, or membrane localization [40,41]. Mesoscale models that predict how clathrin density drives membrane budding have studied productive clathrin assembly, without explicit protein components or time-dependence [42–44]. Nonspatial kinetic models can predict the time-dependence of assembly with multiple clathrin-associated proteins, but have not addressed how stoichiometry or clathrin cooperativity and structure control unstable vs stable lattice assembly on membranes [45,46]. Thus, previous models have not determined, either quantitatively or qualitatively, the minimal criteria for stable nucleation and growth of clathrin lattices on membranes, where it is biologically relevant. Our model here is able to quantitatively define minimal criteria for stable growth vs transient and unstable structures by using recently developed structure-resolved reaction-diffusion software [47], which captures coarse molecular structure [31] for multi-component systems in 3D [48], 2D [49], and transitioning between [50].

In this paper, we first introduce the reaction-diffusion (RD) model, detailing the components present, the fixed kinetic and geometric parameters, and the 'free' parameters that are optimized to obtain agreement with the experimental kinetics. We describe how dimensional reduction (the change in search space and dynamics that accompanies transitions from 3D to 2D [51]) is rigorously accounted for in our model, showing how it quantitatively impacts the stability [19] and kinetics [52] of assembly steps. Using our model optimized against *in vitro* experiments, we can then study clathrin-coat nucleation and growth at a physiologic *V/A* ratio. We vary adaptor density to determine the quantity and stoichiometry of adaptors

necessary to drive efficient and stable lattice formation. Our simulations provide details on each clathrin monomer and higher-order structure as they evolve in time and space. Using concepts from classical nucleation theory, we can thus quantify the critical nuclei as a function of adaptor density for the first time, showing that even with abundant clathrin concentration, a minimum adaptor density is necessary to drive nucleation and continued growth at relevant physiologic timescales (~60s). A tradeoff in our model approach is that we have not dynamically coupled the clathrin lattice assembly with membrane budding. Thus, while we are uniquely able to capture kinetics of localization and assembly on a membrane by tracking flat lattice assembly, which is consistent with observed early growth [53], we study the mechanical energy of membrane bending independently. We note that *in vitro*, clathrin assembles both flat and vesicle-shaped lattices[16–18], just as is observed *in vivo* [53,54]. Specifically, we calculate the energy of a continuum membrane that has reshaped to optimize its bending energy under constraints from pre-assembled curved solution cages. We compare this energetic cost with the energetics one might expect to gain from forming a curved over a flat lattice, as compared to experiment. We discuss connections of our model to other self-assembly systems, and directions for further model development.

## Models and methods

### Reaction-diffusion (RD) simulations

Simulations of all models except the continuum membrane model are performed using the NERDSS reaction-diffusion software [47]. The software propagates stochastic particle-based and structure-resolved RD. NERDSS uses the free-propagator reweighting (FPR) algorithm [48] that is derived for the Smoluchowski model [55] of diffusion with reactive collisions between particles, recovering exact association kinetics and equilibria. The algorithms in NERDSS have been extensively validated for reaction dynamics in 3D [48], 2D [49], from 3D to 2D [50], and with structural resolution and rotational diffusion [31].We use a time-step $\Delta t = 3$ μs. The accuracy and reproducibility of the simulation results is verified using a smaller time-step (Fig A(B) in S1 Text).

To briefly summarize the solver [47], each protein/molecule in the system has a rigid structure composed of point particle coordinates for each binding interface (Fig 1). The system is initially populated with molecules placed randomly within the volume or on the surface. In each time step of the RD simulations, every protein either diffuses (translation and rotation) or undergoes a reaction. Reactions are evaluated first, including 0[th] order creation and 1[st] order destruction, all modeled as Poisson processes. Each bound complex is evaluated for (1[st] order) dissociation. Finally, each pair of reacting interfaces that are within colliding distance are evaluated for binding events. The reaction probabilities for binding events are calculated from the Green's function (GF) solution of the diffusion equation describing the separation $r$ between two particles with a reactive boundary collision occurring at a binding radius of $r = \sigma$, parameterized by an intrinsic rate $k_a$ [48]. The GF and dimensions of the rate $k_a$ depend on the dimensionality of the reaction. If no reaction happens for the members of one complex (the most common outcome given the small time-steps), the whole complex will diffuse as a rigid body. For translational diffusion, the displacements in each dimension are determined via simple Brownian updates, $x(t + \Delta t) = x(t) + \sqrt{2D_x \Delta t}R_x$ where $R_x$ is the standard normal distribution, same in $y$ and $z$, and $D_x$ is the diffusion coefficient in the $x$ direction. For rotational diffusion, the complex rotates around each of the global x, y and z axes by an angle $\theta_i = \sqrt{2D_{Ri}\Delta t}R_i$, with $D_{Ri}$ being the rotational diffusion coefficient around a specific axis.

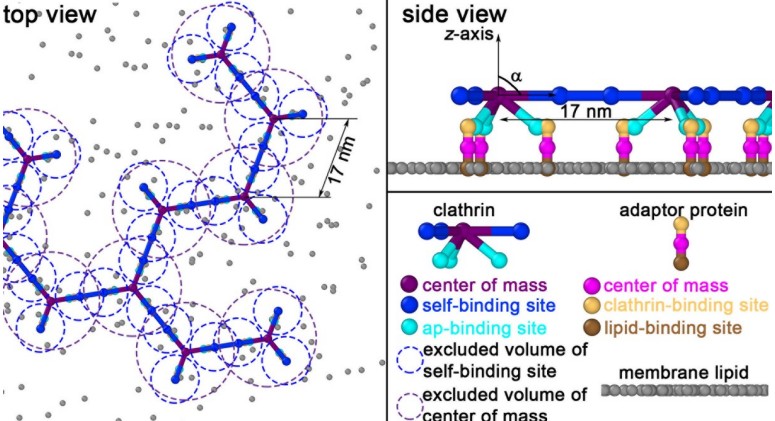

**Fig 1. The structure-resolved reaction-diffusion model for clathrin recruitment and assembly on membranes captures structure, valency, and excluded volume.** Clathrin trimers are initialized in solution, but bind to adaptor proteins that are localized to the surface. Clathrin binds to adaptors only after the adaptors have localized to the surface, which is consistent with the behavior of the adaptor AP-2[16]. Adaptor proteins can bind specific lipids on the membrane surface when initialized in solution. Clathrin-clathrin sites and clathrin-adaptor sites exclude one another with a radius of σ = 5nm (blue circles) and σ = 1nm, respectively, but only when they are unbound; bound sites do not exclude volume. Therefore, at all times in the simulation, center-of-mass (COM) sites on trimers exclude other trimer COMs at σ = 10nm (purple circles), to ensure the lattices do not overlap with one another.

New positions are rejected and resampled if they violate excluded volume of reactant interfaces.

Orientations are enforced only after binding events are accepted between two complexes (or molecules). As characterized in detail in previous work [31], this allows us to directly connect the microscopic parameters, including the rate $k_a$, to the macroscopic rates measured experimentally, $k_{on}$. In 3D for example, $k_{on} = (1/k_a + 1/4\pi\sigma D)^{-1}$, where $D$ is the total diffusion constant due to translation of both molecules, with an effective correction for rotating molecules [31] (see Ref [47] for formula in each dimension). When binding occurs, two complexes are moved to contact at the binding radius σ and "snapped" according to the pre-specified orientation for the reaction from the simulation input. Binding events that cause steric overlap in the system are rejected. We also reject binding events that cause unphysically large rotational motion of the complexes, where we have validated that this constraint does not significantly influence the association kinetics expected from the rate $k_a$ (Fig B in S1 Text and Text A in S1 Text). The binding sites restricted to the membrane are modeled using an implicit lipid algorithm, that reproduces the same kinetics and equilibria as the more expensive explicit lipid method [50]. Software and executable input files for the models used here are provided open source at github.com/mjohn218/NERDSS.

## Model components and structural details

To reproduce the quantitative kinetics of recently measured clathrin assembly on membranes [25], our model captures the coarse structure of the clathrin trimer and its adaptor protein, protein (and multi-protein) diffusion constants, rates describing all pairwise protein-protein interactions, and the concentrations and $V/A$ ratio of the experiment. Each rigid clathrin trimer contains 3 sites for binding other clathrin, arranged to produce a flat or curved hexagonal lattice depending on the 'pucker' angle α of the legs relative to the $z$-axis (Fig 1). Each trimer also contains 3 sites to bind adaptors. Each adaptor contains a single site for clathrin, and a single site to bind a membrane lipid. The pairwise reactions between component interfaces of the

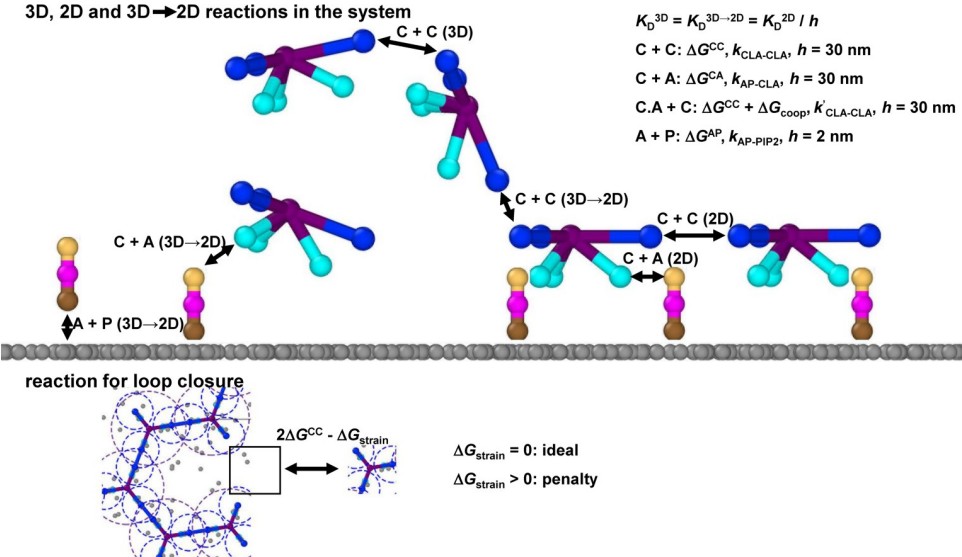

**Fig 2. Reaction network for the model captures energetics and kinetics of binding in 3D, 2D, and between 3D-2D.**
For components clathrin (labeled C), adaptor (A), and lipid (P), only the specific reactions shown in black arrows occur between the connected interfaces. The legend indicates what parameters per reaction are specified, where a 2D reaction relates to the corresponding 3D reaction via a lengthscale $h$ due to the change in dimensionality of the search. When a clathrin-adaptor complex forms (C.A), it binds other clathrin more strongly via $\Delta G_{coop}$. The lower panel shows how the addition of a monomer can close a hexagonal loop. The free energy of forming both bonds is modeled with negative cooperativity, via an energy penalty $\Delta G_{strain}$.

model are summarized in Fig 2. The lengthscales of the proteins are chosen to match the known size of the clathrin lattice [56] with a center-to-center distance of 17nm. The adaptors are modeled as rods, with a binding site that binds to clathrin 4nm above the membrane surface. We model the adaptors both implicitly (for the *in vitro* model) and explicitly (for the *physiologic-like* model). Implicit adaptors do not unbind from the surface, whereas explicit adaptors do. Excluded volume (Fig 1) is essential to prevent clathrin lattices from overlapping as they diffuse in 3D or 2D. Although the FPR algorithm naturally captures excluded volume between reactive binding sites, these sites no longer exclude volume once they are bound. Therefore, we add excluded volume between clathrin COM, which we verified did not alter the kinetics (Fig A(A) in S1 Text). When two molecules bind via their specific interaction sites, they adopt a pre-specified orientation relative to one another. On membranes, the clathrin produces a flat hexagonal lattice free of defects ($\alpha = 90^{\circ}$). In solution, we assemble spherical cages ($\alpha > 90^{\circ}$) composed of hexagons, pentagons and imperfect contacts due to the rigidity of the trimers [47]. Imperfect contacts are still able to form bonds at a specified cutoff of 5.5nm (Text A in S1 Text), which contributes to stabilizing the lattice. In previous work we tested cutoff distances that support physically reasonable lattice structure with minor effects on assembly kinetics [47].

## Transport

We estimate transport properties from the Stokes-Einstein equation, with $D_{Cla} = 13\mu m^2/s$, $D_{R,Cla} = 0.03$ rad$^2$/μs, $D_{ap} = 25\mu m^2/s$, $D_{R,ap} = 0.5$ rad$^2$/μs, $D_{lipid,x,y} = 0.5\mu m^2/s$, $D_{lipid,z} = 0$, and $D_{R,lipid,z} = 0.01$rad$^2$/μs, to allow bound complexes to rotate on the surface. Diffusion slows as complexes grow, consistent with the assumption that the hydrodynamic radius of the complex is the sum over constituents [31]. Specifically, for a complex with $N$ molecules, the transport

coefficients are given by: $D_x = [\sum_{i=1}^{N} D_{xi}^{-1}]^{-1}$ and $D_{Rx} = [\sum_{i=1}^{N} D_{Rxi}^{-1/3}]^{-3}$. For example, adaptor proteins on the membrane have a translational diffusion constant in $x,y$ of 0.49μm$^2$/s.

## Energetic and kinetic parameters

We fixed several energetic parameters based on experimental data, whereas free parameters are in Table 1. We use fixed dissociation constants of $K_D = 120$μM for clathrin-clathrin (without adaptor) [26] and 25μM for clathrin-adaptor [27] binding. It is long known from experiment that the clathrin-clathrin $K_D$ is strengthened when a clathrin is bound to an adaptor [57], but to what extent is not quantified. We therefore introduce a variable, $\Delta G_{coop}$, to capture this. Specifically, using, $K_D = c_0 \exp\left(-\frac{\Delta G}{k_B T}\right)$, where $c_0$ is the standard state concentration (1M), the binding free energy without adaptors bound is $\Delta G_{-ap2} = -9 k_B T$, and with adaptor bound it is $\Delta G_{+ap2} = \Delta G_{-ap2} + \Delta G_{coop}$, where $k_B$ is Boltzmann's constant and $T$ is the temperature. We tested $\Delta G_{coop}$ values in a range -1.1 to -3$k_B T$, after establishing that having $\Delta G_{coop} = 0$ was not able to reproduce the observed kinetics. We keep the clathrin-clathrin off-rate unchanged and accelerate the on-rate by $f_{coop} = \exp(-\Delta G_{coop}/k_B T)$.

Our model accounts for the changes to rates and equilibrium constants that accompany localization to the 2D membrane [58,59], where the model solves for reactions restricted to the surface based on 2D diffusion with 2D rate constants that have distinct units of um$^2$/s [49]. The conversion from 3D to 2D rates is most compactly defined by a molecular length-scale $h$ that is approximately the height range in which the membrane-bound molecule is confined, giving $k_a^{2D} = k_a^{3D}/h$. Because $h$ is fundamentally dependent on the entropic and enthalpic contributions to binding that change following the restriction to the effectively 2D membrane surface, it is dependent on the specific binding pair and thus cannot be exactly predicted without experimental or molecular simulation-based analysis [58,59]. Nonetheless, it is expected to be on the nanometer scale [58]. For 2D interactions involving clathrin, we therefore treat it as a free parameter and tested values of $h = 1$-100nm, the order-of-magnitude of the molecular clathrin length-scale. We assume the microscopic dissociation constants are the same in 3D and 2D, $k_b^{2D} = k_b^{3D}$. The connection between the $K_D$ values and rate constants for each reaction are fixed via $K_D = k_{off}/k_{on} = k_b/k_a$.

We introduce a strain energy $\Delta G_{strain}$ for the formation of closed polygons (hexagons or pentagons) (Fig 2). When the clathrin lattice forms a closed hexagon (or pentagon), dissociation of a single clathrin-clathrin bond does not release a monomer or fragment. The ratio of rebinding to unbinding rate is given by $\frac{k_{rebind}}{k_{unbind}} = \exp\left(-\frac{\Delta G_{CC}}{k_B T}\right) \exp\left(-\frac{\Delta G_{strain}}{k_B T}\right)$, where $\Delta G_{CC}$ is the free energy difference of an unbound to bound pair of clathrin [47]. The term $\exp\left(-\frac{\Delta G_{CC}}{k_B T}\right)$ is simply $\frac{K_D}{C_0}$ for the pairwise reaction, giving a rebinding ratio of 8.3x10$^3$ or 9.2x10$^4$ for clathrin-clathrin bonds without and with adaptor bound, respectively, when

**Table 1. Each parameter varied in the RD model to reproduce *in vitro* experimental kinetics, along with the optimal values we found.** Other energies, geometries, and concentrations were fixed by experimental data.

| Parameters | Explanations | Optimal values |
| --- | --- | --- |
| $\rho_{AP}$ | AP-2 density on membrane | 0.009 nm$^{-2}$ |
| $k_{AP-CLA}$ | AP-2 & clathrin binding rate | 0.0012 μM$^{-1}$s$^{-1}$ |
| $k_{CLA-CLA}$ | Clathrin & clathrin binding rate without AP-2 | 0.083 μM$^{-1}$s$^{-1}$ |
| $\Delta G_{coop}$ | Cooperative energy from AP-2 binding | -2.4 $k_B T$ |
| $\Delta G_{strain}$ | Strain energy for closed polygons | 6.9 $k_B T$ |
| $h$ | Length-scale from dimensional reduction | 30 nm |

$\Delta G_{\text{strain}} = 0$. The much faster rebinding rates render hexagons highly stable against dissociation. The term $\exp\left(-\frac{\Delta G_{\text{strain}}}{k_B T}\right)$ thus lowers the stability of a hexagon compared to 6 ideal bonds (for $\Delta G_{\text{strain}} > 0$). There are several reasons we would expect this because clathrin is a flexible molecule that forms varied lattices and cages. Forming two bonds to close a hexagon/pentagon could place elastic strain on the clathrin, reducing its stability. Some *in vivo* lattices, particularly the flat lattices, also form highly imperfect 'hexagons' that clearly have poorly aligned contacts[60], contributing to a less stable and more dynamic lattice. With a $\Delta G_{\text{strain}}$ value of $+6.9 k_B T$ (this value decreases the binding affinity $10^3$ times for the formation of closed polygons), the free energy of the hexagonal 6-mer corresponds to the strength of 5.4 ideal bonds rather than 6 (e.g. $[6\Delta G_{\text{CC}} + \Delta G_{\text{strain}}]/\Delta G_{\text{CC}}$), whereas a 6-mer in an extended conformation contains 5 bonds.

## Simulations to model in vitro experiments

The experimental $V$ was 200μL [25] and the experimental $A$ of $2.017 \times 10^8$ μm$^2$ is calculated based on 1 nmol of total lipid, with an average lipid SA of 0.7nm$^2$ and a two-leaflet bilayer forming multiple long cylindrical tubules. The experimental $V/A$ ratio is 991μm. Although our simulations fail to capture the membrane topology of individual tubules, by retaining the same surface area to volume and binding sites as the experiments, our simplified flat surface recapitulates the primary role of the membrane as a recruitment site and platform for assembly. Our simulation volume has a flat SA of 1μm$^2$, which would require a height of 991μm. To retain this same $V/A$ ratio without propagating a huge excess of solution clathrin, we instead use a height of 1μm and maintain a constant (stochastically fluctuating) concentration of $[\text{Cla}]_{\text{tot}} =$ 80nM (48 copies) in solution (Fig 3A and see Text A in S1 Text). This assumes the total concentration of clathrin in solution (experimentally set at 80nM [25]) does not change as clathrin accumulates on the membrane, which is true to a very good approximation, with <5% of total clathrin ending up on the membrane by 100 seconds.

   In the *in vitro* experiments, the adaptor proteins were introduced first to the system to coat the membrane, with the unbound adaptors flushed out prior to the addition of clathrin [25]. We therefore model the adaptors as fixed to the membrane, and at time zero there are no clathrin in solution. The clathrin are introduced via a 0$^{\text{th}}$ order reaction and are ultimately maintained at 80nM in solution (Text A in S1 Text). While one expects equilibration of some bound adaptors back to solution, we treat the adaptor density as a variable parameter, as we describe below. The benefit of having them fixed is that it orders-of-magnitude faster for us to simulate them, treating them as 'implicit' binding sites [50] for these *in vitro* simulations. These sites have a height of 4nm, which ensures accurate membrane binding kinetics, as verified by changing the simulation time-step (Fig A(B) in S1 Text) and by a simple system without the clathrin-clathrin reaction (Fig A(C) in S1 Text). The density of adaptors on the membrane prior to clathrin addition was not precisely fixed by the *in vitro* experiments, and thus the density of adaptors ($\rho_{\text{AP}}$) in the *in vitro* simulations are treated as a variable parameter. Specifically, although the density of membrane binding sites was known (0.0746/nm$^2$), we would not expect all these sites to recruit adaptors. For a $K_D$ ranging from 1–10μM [61] between chelator and His-tagged adaptor (present at 200nM) [25], we would expect ~0.0014–0.012/nm$^2$ adaptors on the surface before the clathrin is flowed in, which we use as limits in optimizing $\rho_{\text{AP}}$.

## Optimization

The six parameters that we varied to optimally reproduce experiment are summarized in Table 1. To find optimal parameters, we performed simulations by manually varying each of

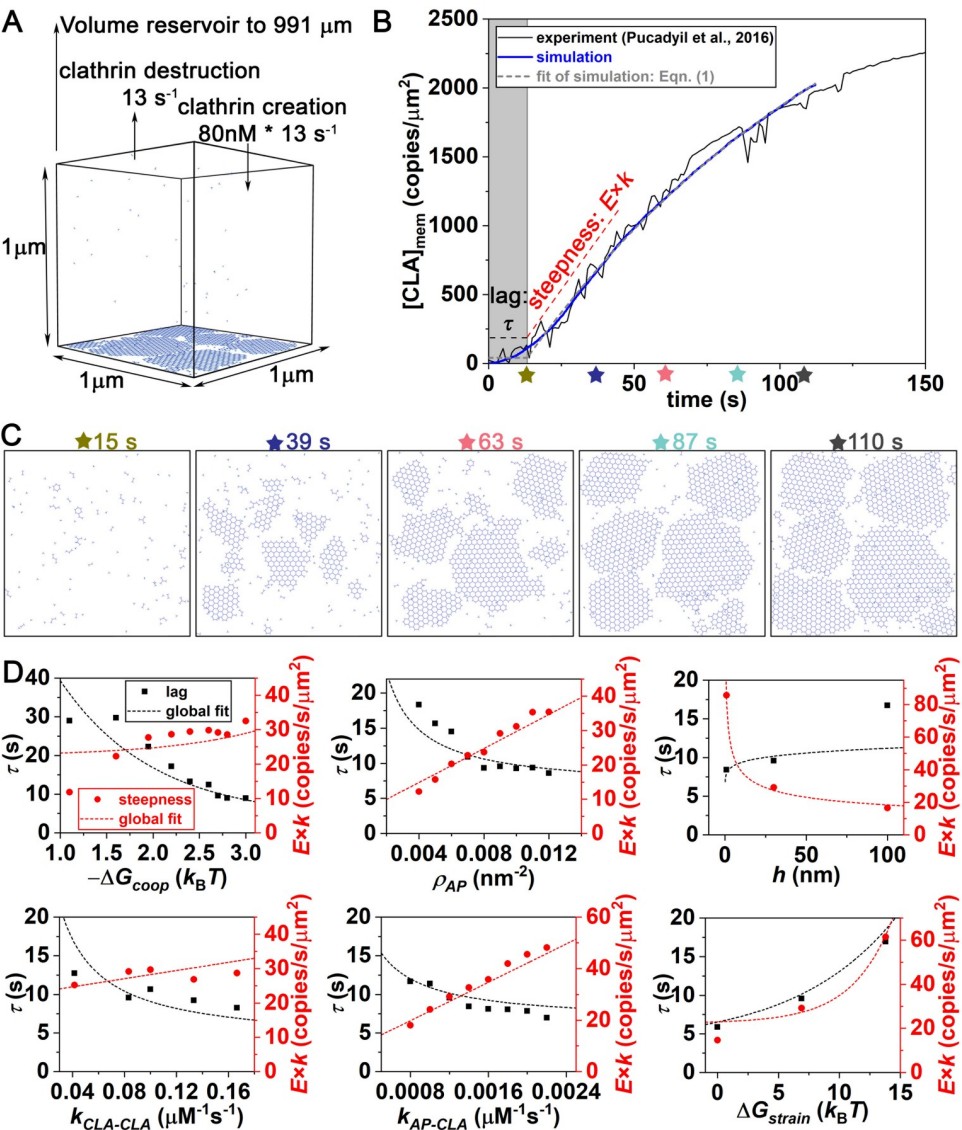

**Fig 3. Structure-resolved reaction-diffusion simulations of clathrin assembly kinetics on membranes reproduce *in vitro* experiment.** (A) Simulations were set up to mimic experimental conditions from published work of Pucadyil *et al* [25], where fluorescence of clathrin accumulating on membrane tubules was measured with time. A constant (fluctuating) solution concentration of clathrin (at 80nM) was maintained through exchange with a large volume reservoir. (B) Fluorescence of clathrin on membranes is averaged over multiple tubules (black line). The experimental fluorescence data was reported in arbitrary units [25]. This means there is a free scale factor we use to match with our simulations, which measure clathrin accumulation in units of copy numbers per μm². The model result is shown in blue (averaged from 4 simulation trajectories). Both the simulation data (gray dashed line) and experimental data can be fit to Eq 1A, producing timescales (growth rate $k$ and lag time $\tau$) that can be directly compared to one another. The initial growth is approximately linear, with a steepness given by $kE$, with $E$ the maximal extent of clathrin on the membrane (Eq 1B). (C) Snapshots of one simulation trajectory at different time points, also see S1 Movie. (D) The macroscopic timescales of the lag time $\tau$ (black dots) and initial growth steepness $kE$ (red dots) vary with changes to six model parameters, as shown in each sub-plot $D_1$-$D_6$ and listed in Table 1. We plot the initial growth steepness $kE$ rather than the growth rate $k$, as it was more amenable to theoretical predictions. The dashed black and red lines are the theoretical fits from our phenomenological models to $\tau$ (Eq 2) and $kE$ (Eq 3). For each subplot, the non-varied parameters are otherwise fixed to the optimal values of Table 1.

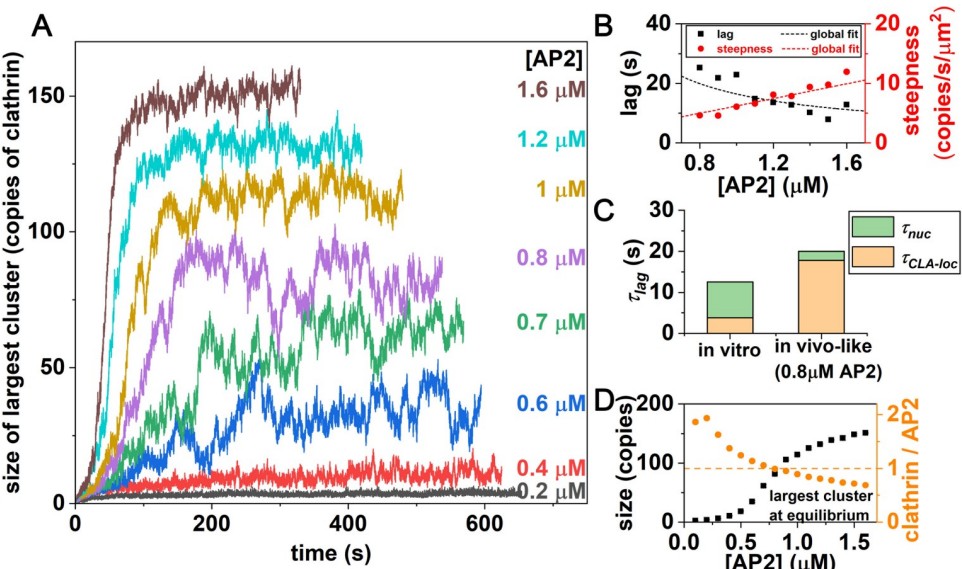

**Fig 4. Clathrin lattices at physiologic-like geometries fail to assemble until adaptor concentration reaches 0.6μM.**
(A) Kinetics of the largest clathrin lattice assembly is faster and reaches larger sizes with increasing adaptor concentration. All simulations have 0.65μM clathrin. (B) Lag times and steepness of initial growth determined from simulation (black and red dots, respectively) follow the theoretical expressions of Eq 2 and Eq 3 relatively well. We excluded the low concentrations where the kinetics is not well-described by the lag and exponential growth model of Eq 1. (C) Lag times are typically slower in these geometrically physiologic-like simulations and rate-limited by localization of clathrin to a membrane (term 1 in Eq 2) with far fewer adaptor binding sites than *in vitro*. (D) The size of the largest lattices increases with increasing adaptor concentration, transitioning from small to large lattices at ~0.8μM (black squares). At this point, the stoichiometry of clathrin:adaptor has reached 1:1 (orange circles).

these parameters in two stages. Initially we performed a broad scan with simultaneous changes to rates, Boltzmann weighted energies, and adaptor densities where each was varied by orders-of-magnitude. While we found the kinetics quite sensitive to the values of $\rho_{AP}$ and $k_{AP-CLA}$, in the broad search we also established the importance of specifying a $\Delta G_{coop}$ value less than zero, and a length-scale $h$ that was >1nm. Specifically, in both stages of optimization, many of the simulated parameter sets could be terminated early on because they showed a very short lag time (<<11s), or showed overly rapid or slow growth. We then performed a more refined search based on this insight, varying parameters again simultaneously across the ranges shown in Fig 3D, where, for example, the rate of clathrin-AP2 binding was sampled from 800M$^{-1}$s$^{-1}$ to 2400M$^{-1}$s$^{-1}$. We chose not to use learning-based or unguided global optimization approaches, as it was straightforward to run many simulations simultaneously along a multi-dimensional grid of parameters. All simulations with kinetics that could be reasonably fit to a model with a lag period followed by exponential growth are reported in S1 Table. The sensitivity of the macroscopic kinetics to each parameter can be relatively reliably estimated by the data and functions shown in Fig 3D (and Fig 4B), as quantified in Eqs 2 and 3.

The experimental data has arbitrary units in the y-axis, which means these values must be rescaled to our data. This therefore also means that only the timescales (which are not in arbitrary units) are relevant to us in optimally reproducing the experimental data. The optimal model we report had the best agreement to both the lag time ($\tau$) and growth rate ($k$) observed experimentally. To plot the experimental data on the same y-axis as our simulations, we zero the y-axis offset (subtracted off 300) and rescaled the height by 1.7. This transformation does not impact the measured lag time or growth rate.

### Simulations to model physiologic-like experiments

In the 'physiologic-like' conditions, we allow the AP-2 protein to reversibly bind the membrane, and prevent this full-length AP-2 from binding to clathrin in solution because binding is experimentally observed to be inhibited prior to membrane localization [16]. The values for clathrin and AP-2 copy numbers in HeLa cells, along with the cell sizes, are taken from Ref [62], compiled from experimental studies [63]. For AP-2, we initially use the concentration of the limiting subunit (0.2μM), and for clathrin trimers, we use copies of clathrin heavy chain divided by 3 (0.65 μM). The $V/A$ ratio for a cytoplasmic volume relative to plasma membrane surface area is ~0.5–2μm [19], and here we set it to 1μm, similar to the value for a HeLa cell[62]. Our simulated $A$ is 0.7x0.7μm, and the height is 1μm. Notably, this means we explicitly capture the $V/A$ ratio and there is no longer a reservoir of solution proteins, so the total copy numbers are fixed; each protein that binds to the membrane depletes the supply remaining in solution, unlike the *in vitro* simulations. With these concentrations, if all of the AP-2 molecules localize to the membrane when present at 0.6 μM, their maximal surface density is 361 copies/μm$^2$, which is much lower than the *in vitro* densities (9000/μm$^2$).

### Clathrin simulations in solution

For solution simulations, the same system set-up as the physiologic-like system is used, except no binding sites are present on the membrane, preventing any localization. Clathrin is present at 0.65 μM, with varying concentration of AP-2. The only modification to the model reactions is that the AP-2 can now bind to the clathrin in solution, as if only the appendage domain (rather than the full auto-inhibited protein) is present. The clathrin is 'puckered' to form spherical cages, using angles of $\alpha = 98°$ (see Fig 1) and a softer curvature of 96°, which forms larger cages.

### Membrane mechanics calculations

For the mechanical membrane calculations, the continuum membrane is represented by a curvature-continuous spline via the subdivision limit surface algorithm as in Ref [64]. The initially flat surface, formed by approximately hexagonal triangles, uses periodic boundary conditions with side length 700 nm, for an area of 490000 nm$^2$. The energy of the membrane is defined by a Helfrich Hamiltonian, as described in Ref [64], and parameterized by a bending modulus of $\kappa = 20k_BT$, and a spontaneous curvature of zero, meaning the surface prefers to be flat. The membrane is then coupled to a rigid clathrin cage (pre-assembled from NERDSS simulations) via a set of harmonic bond potentials, one for each clathrin. For each harmonic bond, a virtual site was placed approximately 9 nm from the clathrin COM (ca. 7.5 nm from the adaptor sites). A distance 2.5 nm closer was also attempted with comparable energetics (within 2%). The distance between the virtual site and the membrane patch was determined as in Ref [50]. The energy of the membrane was then minimized using gradient descent. This nearest point on the membrane, tagged in local surface coordinates (face, and internal triangular coordinates) was saved during 50 "sub-iterations" of the Broyden-Fletcher-Goldfarb-Shanno (BFGS) conjugate-gradient class algorithm. A minimization schedule was found that yielded robust fits. Harmonic force constants were gradually increased by a factor of three, beginning with 0.0003 and stopping with 100 kcal/mol/Angstrom$^2$. The clathrin cage was allowed to shift and rotate during minimization.

## Results

### Our model produces quantitative agreement with *in vitro* kinetics of clathrin accumulation on membranes

We performed simulations mimicking the experimental *in vitro* conditions (Fig 3A), using the model designed to capture the structural (Fig 1), energetic, and kinetic properties (Fig 2 and

Models and Methods) of the clathrin and adaptor protein components. By varying the 6 parameters listed in Table 1, we identified a model that reproduced the time-dependent kinetics observed experimentally (Fig 3B and 3C and S1 Movie). To simplify the comparison of the kinetics of membrane-bound clathrin, $[CLA]_{mem}(t)$, we fit all curves to an exponential growth following a flat lag period accounted for by the Heaviside step function H, which is zero before $\tau$ and one thereafter:

$$[CLA]_{mem}(t) = b + (1 - \exp[-k(t - \tau)])H(t - \tau)E \qquad (1A)$$

$$\approx b + kE(t - \tau)(\tau < t < 0.1k^{-1}). \qquad (1B)$$

There are four fit parameters. The primary two timescales are the lag time $\tau$ and the growth rate $k$, while the maximal extent of clathrin observed to accumulate throughout the numerical simulation is given by $E$, and $b$ allows an offset of clathrin accumulated during the lag-time, as the growth is not completely flat in either simulation or experiment. Our optimal model focuses comparison only on reproducing the two timescales, due to the arbitrary fluorescence units for $b$ and $E$ reported from experiment. Our optimal model (see Table 1 parameters) produces timescales $[\tau, 1/k] = [13s, 121s]$ that are in excellent agreement with those of experiment $[11s, 107s]$. Several other models tested produced worse agreement in kinetics (Fig 3D, Fig C in S1 Text, and S1 Table), where we plot in Fig 3D both $\tau$ and the steepness of the initial growth ($kE$), which is approximately linear, as noted in Eq 1B. The size of the clathrin lattice limits $[CLA]_{mem}$ to ~5000 clathrin/um² (~201nm² per trimer). Due to the large surplus of solution clathrin in both experiment and simulation, saturation of clathrin on the surface can occur, and we find that saturation results primarily from a loss of accessible surface area rather than a loss of adaptor binding sites.

## Cooperativity from both dimensional reduction and adaptor binding are required to reproduce experimental kinetics

The rapid growth in the kinetics following an initial lag of very slow growth is a hallmark of cooperativity. Our model captures two sources of cooperativity. First, experiments have established that once clathrin binds the adaptor proteins, its affinity for other clathrins is increased [57]. Based on our simulations, where the $K_D$ without adaptor is fixed [26], our optimal model predicts that the affinity with adaptor is increased by ~2.4 $k_BT$, leading to faster and more stable binding. For weaker cooperativity, both the lag time and the steepness of growth are too slow (Fig 3D, first graph). Second, once clathrin is localized to the surface, it will exploit the reduced geometric search space of the surface (dimensional reduction) to enhance the stability of its interactions with both clathrin and adaptors [19]. This enhanced stability is guaranteed because the dimensionality factor, $DF = V/(Ah)$, is greater than 1 [19], where $h$ is the length-scale controlling the change to the $K_D$ and $k_{on}$ following 2D localization (Fig 2 and Models and Methods). Our optimal model predicts $h = 30nm$, which is a molecular lengthscale comparable to clathrin's radius, as expected for membrane-localized reactions [58]. A smaller $h$ of 1nm produces much too rapid growth, whereas a larger $h$ of 100nm exhibits much too slow growth (Fig 3D third graph, and Fig C in S1 Text).

## The lag time is primarily controlled by the rate of individual clathrin adsorption to the surface

To better understand how the macroscopic kinetics of clathrin accumulation on the membrane depends on the multiple parameters of the model, we developed approximate theoretical

expressions for the lag time $\tau$ and the steepness of initial growth $kE$. Our expression for the lag time is a sum of three time-scales, the first representing clathrin localization to the membrane, the second nucleation of clathrin-clathrin contacts, and a third term for localization of AP-2 to the membrane, which is nonzero only for the physiological-like simulations (Fig 4). The first two terms describe all simulations, and are given by:

$$\tau_{\text{lag}} = \underbrace{3.2(\rho_{\text{AP}}k_{\text{AP-CLA}}[\text{CLA}]_{\text{bulk}}A)^{-1}}_{\text{localization}} + \underbrace{(f_{\text{coop}}k_{\text{CLA-CLA}}[\text{CLA}]_{\text{bulk}}DF^{1/8.2}exp(-\Delta G_{\text{strain}}/(8.3k_BT)))^{-1}}_{\text{nucleation}} \quad (2)$$

where $f_{\text{coop}} = \exp(-\Delta G_{\text{coop}}/k_BT)$, the dimensionality factor $DF = V/(Ah)$, $[CLA]_{\text{bulk}}$ is the clathrin concentration in the solution, and the other variables are in Table 1. The three time-scales were all developed empirically as inspired by a half-time[65] or mean first passage time [52], varying inversely with *rate* x *concentration* in each term. The localization term is also influenced by the copies of adaptor binding sites on the membrane ($\rho_{\text{AP}}A$), as is clear from our simulation data (Fig 3D second graph, black dots). The nucleation term involves clathrin-clathrin contacts, and we find a nonlinear dependence on the dimensionality factor ($DF$) and the elastic strain of the clathrin lattice ($\Delta G_{\text{strain}}$). This is likely because some clathrins bind directly to clathrin from solution, skipping the cooperative mechanisms provided by dimensional reduction. Nucleation at the surface is also less dependent on the total volume, as it siphons only a small number of clathrin.

Our expression for the lag time has 1 fit parameter in the localization term, two in the nucleation term, and one in the AP-2 localization term (see Eq A in S1 Text). These fit parameters were found by globally fitting the lag time of all simulated models (including Figs 3D and 4), simultaneously against Eq 2 with all model variables included, using nonlinear fitting in MATLAB. These same fit parameters thus describe the kinetics of all the *in vitro* and all the physiologic-like simulations below (Fig D in S1 Text, Text A in S1 Text, and S1 Table). In these *in vitro* simulations, the localization time of clathrin to the membrane contributes about 40% of the lag (first term in Eq 2), and is, as expected, most sensitive to both the rate of clathrin binding to adaptor (Fig 3D fifth graph, and Fig C in S1 Text) and the density of adaptor on the membrane (Fig 3D second graph). Nucleation of a clathrin structure (second term) requires multiple clathrin to localize at one position, which is limited primarily by the low concentration of clathrin (0.08μM).

## Steepness of the initial exponential growth kinetics is controlled primarily by recruitment from adaptors

As with $\tau$, we developed an approximate theoretical expression for the steepness of initial growth, $kE$, which tended to follow much more obvious trends than just the growth rate $k$ alone. This expression is motivated by mass-action kinetics, given that it describes the change in concentration of clathrin on the membrane with time following an initial lag. We again use a sum of three terms, the first captures recruitment of clathrin by adaptors, the second recruitment of clathrin by clathrin, and the third captures recruitment of adaptors to the membrane, which is nonzero only for the physiological-like simulations (Fig 4). The first two terms describe all simulations and are given by:

$$kE = \frac{d[\text{CLA}]_{\text{mem}}}{dt}$$
$$= k_{\text{AP-CLA}}\rho_{\text{AP}}[\text{CLA}]_{\text{bulk}}DF^{1/3.2} + 0.0012f_{\text{coop}}k_{\text{CLA-CLA}}[\text{CLA}]_{\text{bulk}}^2 DF^{1/2.5}exp(\Delta G_{\text{strain}}/(3.3k_BT))V/A \quad (3)$$

This expression shows reasonable agreement to our data (Fig 3D, red dashed curves and Fig 4B). For the *in vitro* simulations, the steepness of growth is most sensitive to the first term

(proportional to $k_{\text{AP–CLA}}$) accounting for recruitment of clathrin to adaptors. Growth is further enhanced cooperatively due to 2D interactions between both clathrin trimer-trimer and clathrin-adaptor contacts. Varying $h = k_{\text{on,3D}}/k_{\text{on,2D}}$ at fixed $V/A$ shows the sensitivity to the *DF* (Fig 3D third graph, and Fig C in S1 Text), indicating that 2D binding is necessary to drive experimental growth speeds.

Counter-intuitively, a higher strain free energy $\Delta G_{\text{strain}}$ slows the lag time but *increases* the growth rate (Fig 3D sixth graph), whereas all other parameters cause correlated slow-downs in lag and growth. Increasing the strain penalty destabilizes closed hexagonal structures, promoting disassembly which slows nucleation ($\tau_{\text{lag}}$), as we would expect. However, the subsequent growth is accelerated, due to increased sampling of dynamic and disordered structures with more 'sticky ends' or free legs (1 free leg/trimer in a hexagon, 1.33 free leg/trimer in extended 6-mer), which recruits additional clathrin more rapidly. The fit of Eq 3 to our data was again performed globally against all simulations, just as for the lag-time, where this expression has 1 fit parameter in the first term, 3 in the second, and 1 in the third term (Fig D in S1 Text, Text A in S1 Text, and S1 Table).

## AP-2 alone is insufficient to nucleate lattices at cellular conditions

To characterize nucleation at more physiologic-like conditions, we reduce the $V/A$ ratio from 991μm, to 1μm, representative of a HeLa cell, and increase the clathrin trimer concentrations to 0.65μM [63] (Models and Methods). We find that stable lattices do not nucleate when we use the physiologic AP-2 concentrations of 0.2μM [63] (Fig 4A). Compared to the *in vitro* system, the significantly reduced $V/A$ results in a much lower density of AP-2 on the membrane (~120μm$^{-2}$, compared to 9000μm$^{-2}$), and space for all clathrin molecules on the surface. Most clathrin assemblies are dimers and monomers, with a stoichiometry of 2 clathrin:1 AP-2 that is too low to induce the cooperativity necessary to nucleate (S2 Movie). By increasing the adaptor concentration while keeping the clathrin concentration fixed, we find that only once we reach 0.7–0.8μM of adaptor do we see stabilization of assemblies $> \sim 50$ clathrin, with kinetics that exhibit both a discernable lag and growth phase (Fig 4A). At this point, the clathrin to adaptor ratio has reached ~1:1 (Fig 4D), and equilibrium fluctuations are also largest at this point, indicating a transition in lattice growth (Fig E in S1 Text). With the increased AP-2, clathrin nucleates larger lattices, with both shorter lag times and faster growth (Fig 4B).

The lag-time has increased significantly at lower concentrations of AP-2 (Fig 4B and 4C). Based on our theoretical expression of Eq 2, the primary reason for a longer lag-time is the first term controlling localization of clathrin to the membrane, which becomes rate-limiting with fewer AP-2 binding sites (Fig 4C). The steepness of initial growth is also significantly slower than *in vitro*, where now only a few clathrin are recruited to a μm$^2$ surface area per second (Fig 4B). Here again the few AP-2 available limits recruitment of clathrin to the surface. The recruitment of clathrin by clathrin has also slowed, due to the sensitivity of the steepness to the *DF* in stabilizing clathrin-clathrin contacts on the surface. We note that the recruitment of AP-2 to lipids is not rate-limiting in these simulations (the third terms in both expressions, see Text A in S1 Text), as PI(4,5)P$_2$ coverage is ~20000μm$^{-2}$ (1%), and the rate of AP-2 to lipid binding is simulated relatively fast, at 0.3s$^{-1}$μM$^{-1}$ (S1 Table).

## Assembly of lattices overcomes an initial barrier, followed by small increases in stability during growth

By tracking the lattice sizes sampled throughout a simulation, we can quantify their relative probability throughout the trajectories in $k_{\text{B}}T$ units, $-\ln P_{\text{obs}}(n; t)$ (Fig 5A). This observable at equilibrium is equivalent to the free energy as a function of lattice sizes, $\frac{G(n)}{k_{\text{B}}T} = -\ln P(n)$

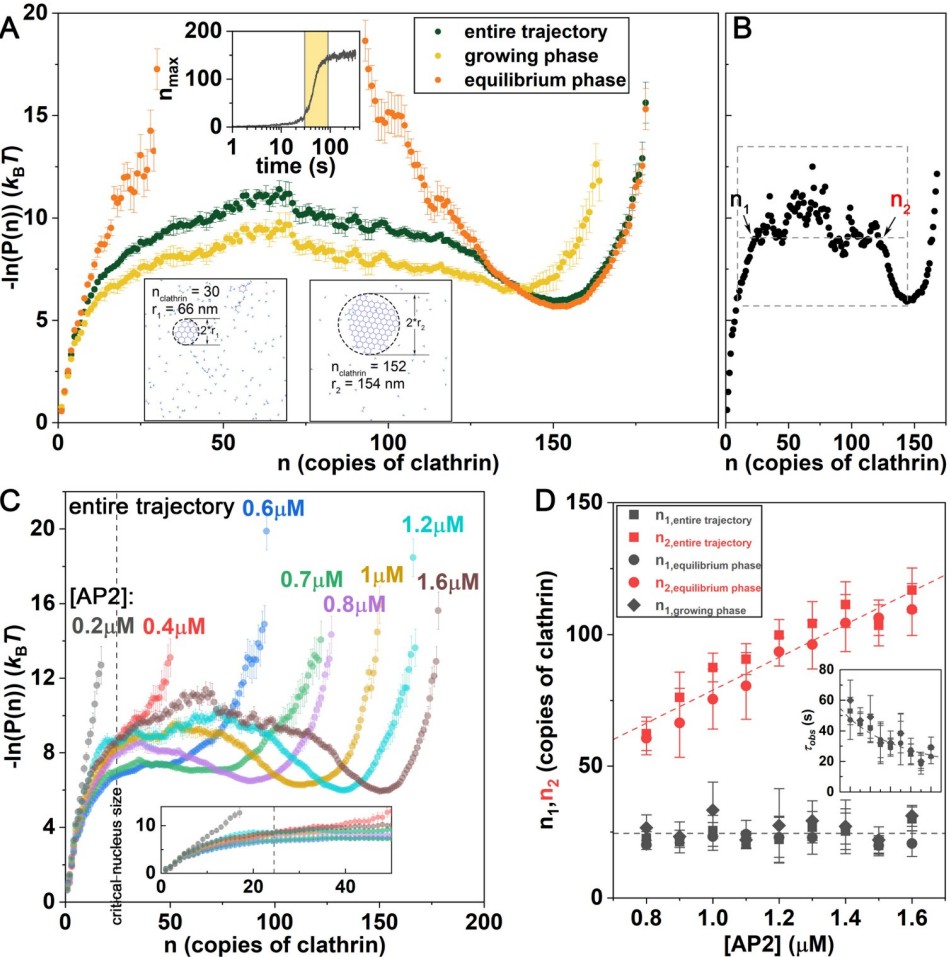

**Fig 5. Clathrin lattices start as monomers that face an initial barrier to growth, with a stable size reached only after significant growth.** (A) The probability of observing clathrin lattices of size $n$ can be converted to an energy-like metric $-\ln(P(n))$, which at equilibrium is a true free energy in units of $k_B T$. An initial barrier (in $n$) to nucleation plateaus, followed by a relatively flat region, where structures have comparable probability. Across the full time-dependent trajectory shown in the upper inset, we analyze lattices sampled across the full trajectory (green dots), the growth phase (yellow dots), and equilibrium (orange dots). During growth, clathrin forms intermediate size lattices, while at equilibrium, only small and large lattices are visible. [AP2] = 1.6μM. (B) To quantify the end of the initial barrier to growth ($n_1$), and the start to the stabilized growth ($n_2$), we define a plateau at constant $-\ln(P(n))$ that defines these intercepts for each trajectory (Text A in S1 Text). (C) With increasing adaptor concentration, larger lattices become stabilized. The shape of the curves are consistent with the data in Fig 4A; the free energy forms a trough at higher concentrations, consistent with the average size of lattices at equilibrium. (D) The critical size where the barrier plateaus is $\sim n_1 = 25$, independent of adaptor concentration. The noisy plateau region is followed by a well that starts at $n_2$ and increases with increasing adaptor concentration. (Inset) With higher adaptor concentration, the time to cross the first barrier at $n_1$ is faster, following the inverse of adaptors $\tau_{obs} \propto 1/[\text{AP2}]$, with a proportionality constant $1/[0.026\mu M^{-1}s^{-1}]$.

(Fig 5A). For higher adaptor concentrations at equilibrium, we observe a bimodal distribution in $P(n)$ with both very large lattices and a population of monomers and small lattices present. This gives rise to a barrier in $-\ln P(n)$ at $n < \sim 25$ (Fig 5A and 5C). This is notable, as it demonstrates that even at higher adaptor concentration, the system does not fully transition into a single coated structure (which would be expected if the free energy monotonically decreased with increasing $n$). After the majority of solution clathrin and adaptor are concentrated into a single coated structure, the remaining and diluted clathrin forms small clusters that are not

stabilized against disassembly. Hence the formation of stable clathrin lattices locally depletes resources needed for nucleation of additional stable sites. The growth phase of assembly (Fig 5A-yellow dots) further shows how even after overcoming the initial barrier to nucleation at small $n$, lattices exhibit similar stability at a broad range of sizes, indicating dynamic remodeling (Fig 5A and 5B). This flat region is followed by a stable well that shifts larger and deepens as the system equilibrates to the maximal lattice sizes, with a final 'wall' that prevents further growth. This wall results from insufficient adaptors to stabilize growth at the peripheries of the lattice, leaving behind a dilute phase of clathrin.

## Adaptor concentration controls the time to reach the observed nucleus, but not its size

As we increase adaptor concentration beyond 0.7μM, we observe that across the full trajectory, the initial barrier to nucleation transitions to a flattened region at a similar lattice size of $n_1 \sim 25$ clathrin for all adaptor concentrations (Fig 5C and 5D). This initial barrier is significant because smaller lattices are prone to disassembly (Fig E(C) in S1 Text). There is a later transition where the stable well starts to form, at $n_2$, which clearly shifts to larger lattices as the adaptor concentration increases (Fig 5C). These two lattice sizes, which bound the noisy plateau region that is sampled during growth, are quite consistent with the lattices sizes that define the equilibrium regions (Fig 5D). We do observe some features in the noisy plateau region across the full trajectory, where a small barrier can exist at higher lattice size ($n \sim 40$–$70$). This indicates that during growth, lattices that are about halfway to the stable size are the least commonly observed, and indeed have a short lifetime (Fig E in S1 Text). Additional adaptors also dramatically accelerate the timescale required to reach the 'critical' nucleus $n_1$, with a time that is approximately proportional to the inverse adaptor concentration (Fig 5D-inset). We note that it is somewhat surprising that the barrier to nucleation $n_1$ is largely independent of adaptor concentration. However, the clathrin concentration in all simulations is the same, suggesting that the barrier is largely encoded by the clathrin concentration and lattice structures formed, with the critical nucleus requiring sufficient clathrin cross-links to form and adaptors only able to bias towards assembling these structures. Indeed, with changes to clathrin concentration, the size of initially stable nuclei $n_1$ does change, indicating that it is the total clathrin available that controls the initial barrier to nucleation (Fig F in S1 Text).

The size of our observed 'critical' nuclei shows that even after multiple (8–10) hexagons (Fig 5A inset) have formed, the lattice can still spontaneously disassemble (see S2 Movie). In particular, at timescales less than the time it takes to reach this critical size, which even at 1.6μM adaptor is ~30s, lattices show a significant amount of dynamic remodeling. Most of the growth and shrinking occurs through monomers (Fig E(B) in S1 Text), and when a lattice does change size, the probabilities to dissociate a trimer or add one are comparable before reaching the maximal size limit (Fig E(C) in S1 Text).

## Assembly in solution is less cooperative than on the membrane, despite increased stability of the curved structures

The above simulations tracked flat clathrin lattices on a flat surface. However, in solution cages are curved [57], indicating that curved cages are more stable, absent the external constraints of the membrane. Recent experiments also quantified that clathrin assembly is more stable onto more highly curved surfaces, illustrating the energy gain that accompanies curved clathrin contacts [66]. Therefore, a curved lattice should have stronger energetics and stability than the flat lattice we simulated on the surface. First, however, we simulated the identical model from the membrane simulations (Fig 4A) in a solution setting with adaptors and 'puckered' clathrin

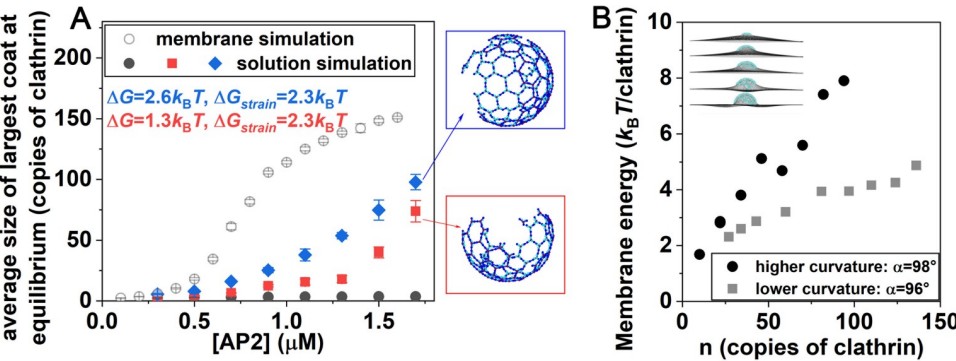

**Fig 6. Curved-cage assembly in solution requires added stabilization, which is comparable to membrane bending energies.** (A) Clathrin cages with adaptors present do not form in solution (black dots), using the same model as on the membrane (open circles), despite the same concentration of clathrin. With added stabilization for all clathrin-clathrin bonds ($\Delta G$) and reduced strain ($\Delta G_{strain}$), solutions cages start to form (red and blue), although growth is less cooperative without dimensional reduction following membrane localization. (B) Membrane bending energy per clathrin trimer as the sizes $n$ of curved lattices are increased. More highly curved cages ($\alpha = 98°$) cost more energy per trimer to bend the membrane. The membrane energy is proportional to the bending modulus $\kappa$, where we use $\kappa = 20$ $k_{B}T$ consistent with measurements on the plasma membrane [67].

monomers (Models and Methods), observing almost no assembly even with high concentrations of adaptors (Fig 6A). This dramatic drop-off in assembly is due to the lack of dimensional reduction in the solution assembly, which stabilizes assembly in 2D.

However, it is known that clathrin cages can form in solution when adaptor peptides like from AP-2 are present, at comparable concentrations to those studied here [57]. Therefore, and consistent with experiments showing the curved cages are more stable [66], we then increased the binding free energy of the clathrin-clathrin contacts. We also reduced the strain penalty on hexagons and pentagons, and with these two energy-stabilizing changes, we see a growth in assembly (Fig 6A). This assembly yield is dependent on the adaptor concentration, which is consistent with *in vitro* experiments on adaptor-peptide driven cage formation in solution [57]; by stabilizing the bonds at ~$2k_{B}T$, and reducing the hexagonal strain by $2.3k_{B}T$, our cage assembly agrees relatively well with *in vitro* experiments (Fig G in S1 Text) [57]. As a result of both bond and hexagonal strain, we estimate that the curved lattice has an additional 3–4 $k_{B}T$ free energy per trimer (Table A in S1 Text) available to perform work on the flat membrane, that is, to bend it. We refer to this quantity subsequently as the "clathrin curvature free energy".

## Membrane bending cost can be at least partially compensated by stability of curved lattice

We now ask if energy gained by moving the lattice from flat to curved on the membrane, which we estimated above was ~3–4 $k_{B}T$ per trimer, could offset the cost of bending the membrane. We measure the cost of bending the membrane by harmonically coupling our pre-assembled clathrin cages with a deformable membrane model [64] that follows the Helfrich Hamiltonian [68] (Fig 6B insets) (see Models and Methods). We find that as the clathrin lattices grow, the bending energy cost per trimer increases, with less energy required for lower curvature lattices (Fig 6B). We see that for the lower curvature lattices in particular, the bending energy cost per clathrin trimer is ~2–4 $k_{B}T$, which is indeed comparable to the "clathrin curvature free energy" gain of 3–4 $k_{B}T$. This result suggests that the cost of bending the membrane by a clathrin lattice can be at least in part driven by the energy gained by forming a

curved clathrin assembly. As we showed already in Figs 4 and 5, there is sufficient energy in the flat clathrin lattice model to assemble as long as enough adaptors are present.

To summarize this point, when the clathrin forms a flat lattice on the membrane, the clathrin lattice is strained and pays a cost (compared to the curved lattice), while the membrane is at rest. When a curved clathrin cage forms, the strain on the lattice is eased, improving its energetics, but the cost of the membrane bending increases. We note that we do not expect the energy of the curved, membrane bound clathrin lattice to recover the stability of the curved solution cages, due to forces placed on it by the deformed membrane. Thus, modeling a flat lattice on the membrane can be, at least energetically, comparable to modeling a curved lattice with stronger stability that is offset by the bent membrane. Modified energetics of clathrin assemblies on membranes compared to solution were also measured in a coarse-grained model of clathrin [33]. We suggest, therefore, that a model that couples assembly with membrane remodeling in a dynamic way could recover similar energies for the curved lattice under bending strain than the flat lattice on flat membranes.

## Discussion

Our model predicts that nascent clathrin lattices on membranes are unstable until ~25 clathrin are present, given physiologic parameters of 0.65μM clathrin and $V/A = 1μm$. Our estimate of the stability threshold is remarkably similar to the sizes of abortive structures in cells [3]. A minimal requirement to overcome this barrier to nucleate structures on a membrane is a sufficient concentration of adaptor proteins that can link clathrin to the membrane, allowing assembly to cooperatively benefit from both increased (adaptor-driven) stability and dimensional reduction. Physiological concentrations of AP-2 alone are insufficient to recruit enough clathrin to make membrane-bound cage formation spontaneous. Therefore, the participation of other adaptors is essential to drive nucleation. Our model shows how *in vitro* conditions with *in vivo* protein concentrations do not sample the transition between unstable and stable lattice growth, in large part due to an unphysiological $V/A$ ratio. Our kinetic simulations and theoretical analysis allow us to bridge the geometric divide between *in vitro* and *in vivo*; our model predicts that for one adaptor type to drive productive nucleation and growth in ~60s (average time for vesicle formation) requires at least 1.6μM adaptors. This also requires sufficient lipid populations that can trigger efficient binding to the membrane, with slower lipid binding rates or reduced populations ultimately slowing both lag and growth rates. Our results clearly demonstrate that spontaneous disassembly of clathrin-coated structures on membranes can occur naturally even after relatively long time periods (many seconds). As evidenced by our equilibrium results, the accumulation of the majority of clathrin into single structures also depletes local clathrin and adaptor populations, leaving remaining clathrin capable of forming only small and transient structures (Fig 5).

By modeling only a single type of adaptor for recruitment of clathrin to the membrane, we were able to construct approximate theoretical formulas to predict how the microscopic variables change the macroscopic lag time and steepness of growth, even as we varied the clathrin concentration, lipid concentration, system geometry, changes in cooperativity (Fig H in S1 Text) or rates of lipid binding (Fig I in S1 Text). However, *in vivo* there are multiple types of adaptors or accessory proteins that can link clathrin to the membrane [69]. Although our model predicts how adaptors with faster or slower binding kinetics would alter macroscopic growth, the ability of many of these proteins to bind to one another [62] creates cross-linking that will stabilize these proteins on the membrane surface [19]. By increasing the local density of clathrin recruitment sites, we predict this cross-linking would then help nucleate stable lattices at lower concentrations than occur for a single adaptor type. The binding kinetics of

additional interactions can also be significantly more dynamic than AP-2; the proteins FCho1 and eps15 form clusters with AP-2 that helps initiate sites of clathrin-coated structure formation in cells, and yet both FCHo1 and eps15 are largely absent from completed vesicles, indicating the transience of their clathrin contacts [70–72]. The behavior of these additional adaptor protein types is consistent with our model prediction that AP-2 needs help to nucleate stable lattices, and in future work we will consider how explicit cross-linking can enhance local concentrations to drive nucleation and growth in time and space. Importantly, the model and software are open source at github.com/mjohn218/NERDSS, so they can be immediately applied and/or modified with additional experimental data (Models and Methods).

A limitation of our model is the clathrin-coat assembly is not dynamically coupled to the membrane remodeling to form spherical vesicles. However, our energetic calculations using our assembled clathrin structures and a deformable membrane model demonstrate that the energy per clathrin needed to bend the membrane is often comparable to the clathrin curvature free energy. Our model thus predicts that the energy of flat lattices on membranes and curved cages on bent membranes is of a similar scale, albeit sensitive to the membrane stiffness and additional curvature generation mechanisms. Additional curvature induction driven by adaptor proteins (e.g. amphipathic helix insertion and BAR domains [73]) would help remodel the membrane, thus reducing the work required from the clathrin lattice alone. The kinetics of membrane bending is unlikely to be rate-limiting, given the slow time-scales of clathrin recruitment and relatively fast remodeling dynamics observed in membrane budding [74]. The kinetics of late cage assembly could be altered due to budding, however, due to the reduction in clathrin lattice perimeter in a curved cage as the size proceeds beyond the vesicle diameter (~150nm). While this would not impact the early growth to overcome the nucleation barrier, the closing of the vesicle involves a shrinking perimeter that reduces sites to recruit clathrin, but increases contacts formed on average per clathrin. Modeling the kinetics of coupled assembly and remodeling will ultimately be important to isolate productive vesicle forming events, and is feasible for the types of models used here.

The clathrin coat assembly studied here shares many fundamental properties with diverse biological pathways including steps in viral assembly [74–76] and protein aggregation [77], where we would expect similar sensitivity to membrane or surface (air/water) localization and geometry. Our work here predicts how assembly and aggregation can occur on surfaces with weaker interaction energies or significantly lower copy numbers than is needed to nucleate structures in solution. Localization to the surface provides an additional time-scale that can slow nucleation, which would be helpful in avoiding kinetic traps [78] to improve yield in designed systems [79,80]. The addition of the membrane provides a control variable for experimental quantification of interaction kinetics, where we showed here how visualization of recruitment to surfaces [25] can be an effective observable for discriminating assembly mechanisms. Enzymatic control of membrane composition can then be used to trigger disassembly, to study unbinding rates [47]. Overall, the rate-based approach used here [47] offers a valuable platform for mechanistic and predictive modeling of self-assembly, in and out-of-equilibrium.

## Supporting information

**S1 Text.** Text A: Supplemental Methods. Table A: Calculations of average energies per trimers assembled into lattices. Fig A: Numerical confirmation of expected kinetics. Fig B: Evaluating how association events that are rejected for producing significant orientational displacements affect reaction kinetics. Fig C: Kinetics of clathrin accumulation on membranes with changing model parameters. Fig D: Correlations between observed lag-time and steepness of initial growth from simulations vs predicted values from the simplified models of Eq 2 and Eq 3 of

main text. Fig E: Growth mechanisms of clathrin lattices at the more "physiologic-like" geometries. Fig F: Growth of lattices with changing clathrin concentration, at fixed 1μM of adaptors. Fig G: Fraction of assembled clathrin in solution vs adaptor concentration. Fig H: Maximal lattices formed as AP2 concentration is varied, comparing two values of cooperativity in clathrin-clathrin contacts, $\Delta G_{coop}$. Fig I: Reduced lipid populations or slower lipid binding rates slow the lag.
(PDF)

**S1 Movie. Video of the clathrin assembly matching the *in vitro* simulations, as shown in Fig 3 of main text.**
(MP4)

**S2 Movie. Videos of clathrin assembly at physiologic-like conditions, as a function of varying adaptor concentration (Fig 4 of main text).**
(MP4)

**S1 Table. Excel sheet containing the parameters for all simulations with discernable lag times and growth rates.** Includes the corresponding observed lag times and steepness of initial growth from simulations, and predicted lag times and steepness from our theoretical expressions.
(XLSX)

## Acknowledgments

We acknowledge use of the ARCH supercomputers bluecrab (MARCC) and rockfish at Johns Hopkins, with support from NSF MRI 1920103 and the XSEDE supercomputer Stampede2 through XRAC MCB150059. We are grateful to Prof Thomas Pucadyil for sharing his data and for helpful discussions on the experiments.

## Author Contributions

**Conceptualization:** Si-Kao Guo, Margaret E. Johnson.

**Data curation:** Si-Kao Guo, Margaret E. Johnson.

**Formal analysis:** Si-Kao Guo, Alexander J. Sodt, Margaret E. Johnson.

**Funding acquisition:** Margaret E. Johnson.

**Investigation:** Si-Kao Guo, Margaret E. Johnson.

**Methodology:** Si-Kao Guo, Alexander J. Sodt, Margaret E. Johnson.

**Project administration:** Si-Kao Guo, Margaret E. Johnson.

**Resources:** Margaret E. Johnson.

**Software:** Si-Kao Guo, Margaret E. Johnson.

**Supervision:** Margaret E. Johnson.

**Validation:** Si-Kao Guo, Margaret E. Johnson.

**Visualization:** Si-Kao Guo.

**Writing – original draft:** Si-Kao Guo, Margaret E. Johnson.

**Writing – review & editing:** Si-Kao Guo, Alexander J. Sodt, Margaret E. Johnson.

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
