## [Decision Letter · Decision Letter 0]

11 Dec 2021

Dear Prof Johnson,

Thank you very much for submitting your manuscript "Large self-assembled clathrin lattices spontaneously disassemble without sufficient adaptor proteins" for consideration at PLOS Computational Biology.

As with all papers reviewed by the journal, your manuscript was reviewed by members of the editorial board and by several independent reviewers. In light of the reviews (below this email), we would like to invite the resubmission of a significantly-revised version that takes into account the reviewers' comments.

We cannot make any decision about publication until we have seen the revised manuscript and your response to the reviewers' comments. Your revised manuscript is also likely to be sent to reviewers for further evaluation.

Sincerely,

Peter M Kasson

Associate Editor

PLOS Computational Biology

Jason Haugh

Deputy Editor

PLOS Computational Biology

Reviewer's Responses to Questions

**Comments to the Authors:**

Reviewer #1: Guo, Sodt and Johnson present simulations of clathrin assembly at the cell membrane. Their simulations using NERDSS reproduce the experimental curves by Pucadyil and Holkar. While I agree with the authors that simulations provide a powerful tool to help understanding the experiments, I do not think this study is there yet.

Ideally a study should be self-contained, which admittedly is difficult when using a complex model. The level of information provided in this manuscript is too low: it should, e.g., not be necessary for a reader to guess that the simulations are implemented as 1st order Brownian Dynamics.

The rotational diffusion coefficient of the triskelion, at 0.03 rad^2/sec, is a million times smaller than the value reported in the literature based on hydrodynamic calculations. Many triskelia will not even have rotated 180 degrees during the simulation of ~100 seconds.

Experimental binding constants can only be plugged directly into the simulation when proteins are modeled as structureless particles. The directionality in the current model -- proteins only bind when they are positioned and oriented approximately correctly -- requires an adjusted reaction constant to reproduce the experimental binding constant.

Binding of the triskelia to the membrane via AP2 alters the oriental distribution of these bound triskelia relative to the isotropic distribution in solution, and thereby affects their binding into aggregates. This change should not be introduced artificially with the tuning parameter $h$, as it is already automatically generated by the non-spherical protein model.

What is the physical reality of the 'initial barrier to growth'? The free energy barrier to lattice formation is not lowered by the system being out of equilibrium.

In Eqs [Disp-formula pcbi.1009969.e014] and [Disp-formula pcbi.1009969.e015], why has the temperature (k_B T) become a fit parameter? Why do only two out of three terms acquire a fitted proportionality constant?

Minor issues

* The second movie would benefit from drawing the triskelia with thicker lines.

* How is E defined; is is the theoretical maximum or the actual maximum achieved in the simulations?

* Fig 3 is referred to before Fig 2, Fig 6 before 2 through 5, etc.

Reviewer #2: The authors study the assembly of clathrin-coated structures on lipid membranes, using a structure-resolved reaction diffusion equation solver framework that they have developed. They particularly focus on the poorly understood experimental observation that the critical nucleus size of such clatherin-coated structures is surprisingly large, containing on the order of 20 or more clathrin subunits. Through extensive modeling, they determine factors that control the stability of clathrin-structure intermediates; in particular that excess adapter protein is crucial for enabling nucleation.

The article is for the most part thoroughly and clearly written, the modeling is comprehensively described, and the relevance of the results is strongly supported by comparisons against recent in vitro experimental measurements.

The manuscript also provides a good description of previous work, placing the current work in the proper context.

The simulation results provide a detailed understanding of how clathrin assembly depends on relevant control parameters, and the prediction that excess adapter concentrations are required for nucleation is an important observation which is unlikely could be made through experiments alone.

I have a few minor comments on the text:

-for the benefit of readers with limited cell biology knowledge, the authors might include an additional sentence or two about why understanding clathrin assembly is biologically important.

-The description of the reaction-diffusion equation solver framework could be described in a bit more detail, although it is noted that these descriptions are in prior works.

- page 9, “When two molecules bind via their specific interaction sites, they adopt a pre-specified orientation relative to one another.”

Can the authors elaborate on this? Does this mean that when two interaction sites come into contact, they immediately reorient into some desired orientation, regardless of the contact angle? How does this affect association kinetics?

-Relatedly, can the authors describe more clearly the limitations of the algorithm; for example, to what extent does the model account for incorrectly oriented clathrin-clatherin interactions and how might this affect results.

-Can the authors describe in more detail the motivation for setting \\Delta G_strain=6.9? It is not clear how the authors settled on such a precise value.

-A positive aspect of the modeling is that many of the parameters are fixed by experiment. This does leave six parameters to be optimized. From the description given in the text, the optimization procedure seems rather unsystematic. Can the authors provide estimates of sensitivity of the results to optimized parameter values, or the extent to which the fit parameter values are globally optimized?

-“Transport We estimate transport properties from Einstein-Stokes, with DCla=13m2/s,

DR,Cla=0.03 rad2/s, Dap=25m2/s, DR,ap=0.5 rad2/s, Dlipid = 0.5m2/s, and DR,lipis=0.01rad2/s, to

allow bound complexes to rotate on the surface. Diffusion slows as complexes grow, consistent

with Einstein-Stokes29. For example, adaptor proteins on the membrane have a translational

diffusion constant of 0.49m2/s.”

Can the authors elaborate on this? Where do these diffusion constant values come from? What does it mean that “Diffusion slows as complexes grow, consistent with Einstein-Stokes”? Is this based on hydrodynamic radius of a complex?

- p24: “For higher adaptor concentrations at equilibrium, we observe a bimodal distribution in ()… This is a notable outcome, as … after the majority of solution clathrin and adaptor

are concentrated into a single coated structure, the remaining clathrin forms small clusters that

are not stabilized against disassembly”

Why is this a notable outcome? It sounds like you are just saying that at equilibrium the system undergoes macroscopic phase separation, as one would expect given that there is nothing in the model that would stop cluster free energies from monotonically decreasing with size after the initial barrier is crossed.

-Fig. 5 and related text: the manuscript seems to say that the barrier to assembly ends around size n \\approx 25, and is then followed by a relatively flat but noisy free energy profile, regardless of parameter values. Is this correct? Normally from classical nucleation theory the critical nucleus size would depend on binding affinity values and subunit concentrations. Can the authors explain why that is not so much the case here?

**Have the authors made all data and (if applicable) computational code underlying the findings in their manuscript fully available?**

Reviewer #1: Yes

Reviewer #2: Yes

PLOS authors have the option to publish the peer review history of their article (what does this mean?). If published, this will include your full peer review and any attached files.

Reviewer #1: No

Reviewer #2: No
---

## [Decision Letter · Decision Letter 1]

24 Feb 2022

Dear Prof Johnson,

We are pleased to inform you that your manuscript 'Large self-assembled clathrin lattices spontaneously disassemble without sufficient adaptor proteins' has been provisionally accepted for publication in PLOS Computational Biology.  Both reviewers recommend acceptance; one has three small changes requested that can be accomplished along with the formatting changes for publication.

IMPORTANT: The editorial review process is now complete. PLOS will only permit corrections to spelling, formatting or significant scientific errors from this point onwards with the exception of the three items above. Requests for major changes, or any which affect the scientific understanding of your work, will cause delays to the publication date of your manuscript.

Best regards,

Peter M Kasson

Associate Editor

PLOS Computational Biology

Jason Haugh

Deputy Editor

PLOS Computational Biology

Reviewer's Responses to Questions

**Comments to the Authors:**

Reviewer #1: The manuscript has improved considerably in the revision.

I recommend publication.

Minor:

p 21: Please add brackets to ( V / A ) / h (twice).

p 26: ... a bimodel distribution with either very large lattices or a population of monomers ... Since both exist simultaneously, either -> both & or -> and.

p30: The bending energy cost is ~2-4 kB T. Please add that this is the total converted by an area term into the cost per clathrin.

Reviewer #2: In my view, the authors have responded adequately to all of the reviewer comments. It's a very interesting study, and in my view the manuscript is ready for publication.

**Have the authors made all data and (if applicable) computational code underlying the findings in their manuscript fully available?**

Reviewer #1: Yes

Reviewer #2: None

PLOS authors have the option to publish the peer review history of their article (what does this mean?). If published, this will include your full peer review and any attached files.

Reviewer #1: No

Reviewer #2: No

---

## [Editor Report · Acceptance letter]

17 Mar 2022

PCOMPBIOL-D-21-01926R1 

Large self-assembled clathrin lattices spontaneously disassemble without sufficient adaptor proteins

Dear Dr Johnson,

I am pleased to inform you that your manuscript has been formally accepted for publication in PLOS Computational Biology. Your manuscript is now with our production department and you will be notified of the publication date in due course.

With kind regards,

Orsolya Voros
